# Investigating the potential of packed cell volume for deducing hemoglobin: Cholistani camels in perspective

Umer Farooq[1]☯, Musadiq Idris[1]☯, Nouman Sajjad[1]☯, Mushtaq Hussain Lashari🆔[2]☯*, Shahbaz Ahmad[2]☯, Zia Ur Rehman[1]☯, Haroon Rashid[1]☯, Aisha Mahmood[1]☯, Sajid Hameed[3]☯

1 Department of Physiology, The Islamia University of Bahawalpur, Bahawalpur, Pakistan, 2 Department of Zoology, The Islamia University of Bahawalpur, Bahawalpur, Pakistan, 3 Department of Anatomy and Histology, The Islamia University of Bahawalpur, Bahawalpur, Pakistan

☯ These authors contributed equally to this work.
* mushtaq.hussain@iub.edu.pk

**Data Availability Statement:** All relevant data are within the paper.

## Abstract

In human medical practice, a hematological rule of three has been validated for healthy human populations. One such formula is estimating hemoglobin (Hb) levels as 1/3$^{rd}$ of Packed Cell Volume (PCV). However, no such hematological formulae have been devised and validated for veterinary medical practice. The present study was devised with an aim to evaluate the relationship between hemoglobin (Hb) concentration and Packed Cell Volume (PCV) in camels (n = 215) being reared under pastoralism, and to devise a simple pen-side hematological formula for estimation of Hb from PCV. The PCV was determined through microhematocrit method whereas Hb estimation by cyanmethaemoglobin method (HbD). The Hb was also calculated as 1/3$^{rd}$ of PCV and was dubbed as calculated Hb (HbC). Overall HbD and HbC were significantly (P≥0.05) different. Similar results were attained for all study groups *i.e.* males (n = 94) and females (n = 121), and young (n = 85) and adult (n = 130) camels. The corrected Hb (CHb) was deduced through regression prediction equation attained from linear regression model. Scatterplots were drawn, linear regression was carried out, and Bland Altman chart was built for agreement of both methods of Hb estimation. A non-significant (P≥0.05) difference was noticed between HbD and CHb. Bland Altman agreement analysis revealed satisfactory agreement between HbD and CHb and the data was distributed closely around the mean difference line (Mean = 0.1436, 95% CI = 3.00, -2.72). A simplified pen-side hematological formula for deducing Hb concentration from PCV is accordingly recommended *viz.* Hb concentration (g/dL) = 0.18(PCV)+5.4 for all age and gender groups of camels instead of its calculation as one-third of PCV.

## Introduction

Packed Cell Volume (PCV), also well-known as hematocrit or erythrocyte volume fraction, is fraction of red blood cells (RBCs) in the animal's blood [1]. It is responsible for transportation

**Funding:** The said research work was funded by the Pakistan Science Foundation (IUB/PSF-931) which has been mentioned online. Furthermore, Nouman Sajjad, one of the coauthors received salary from the said project, please.

**Competing interests:** I have read the journal's policy and the authors of this manuscript have the following competing interests it will helpful to manage the health of animals, and also helpful for professionals This does not alter our adherence to PLOS ONE policies on sharing data and materials.

of oxygen and absorbed nutrients [2]. Amplified PCV not only results in a better transportation but also an augmented primary and secondary polycythemia [3]. Moreover, a high PCV reading pointed out either an increased number of RBCs or decreased volume of circulating plasma. PCV is the most precise way of estimating erythrocyte volume and may also be used to assume total blood volume and hemoglobin (Hb) level. The manual, spun PCV (through microhematocrit method) is a key measurement, underpinning much of hematology. The calibration of virtually all hematology autoanalyzers can be traced in some way back to the PCV [4]. Reference ranges for the hematocrit and red cell indices depend on the validity of this calibration, as do the assignment of expected values to calibrators and controls, and the assignment of target values for statistical population-based quality control programs. Any errors in PCV assignment have far-reaching implications [5].

Extensive research work has been conducted in human medical sciences directed towards assessing an interrelationship between PCV and Hb, and confirming the thumb rule of Hb being 1/3$^{rd}$ of PCV. Certain studies have nullified this rule claiming that Hb estimates cannot be obtained from PCV values with a reliable precision by making use of the common rule of dividing by three [6–8]. The results of these studies also indicated that the association between Hb and PCV is not exactly three times and the sex and age of the individuals can also have a significant effect on this three-fold conversion.

For veterinary medical sciences, the interrelationship between PCV and Hb has been studied for indigenous cattle [9]. In this study assessing Hb as 1/3$^{rd}$ of PCV has been nullified and a newer alternative formula has been reported for Hb assessment through PCV. Similarly, our laboratory has reported similar reports and pen-side hematological formulae for Hb estimation for goats [10] and Cholistani breed of cattle [11]. However, study on such interrelationship for the blood of camels has not yet been reported. The present novel study has therefor been devised with an aim of evaluating the relationship between Hb concentration and PCV in Cholistani camels being reared under pastoralism in Cholistan desert of Pakistan. Furthermore, it also aims to devise a simple pen-side hematological formula for estimation of Hb from PCV.

## Materials and methods

### Geo-location of the study

The study was simultaneously conducted at Cholistan desert, Pakistan (for field blood sampling) and Physiology post-graduate lab of the Department of Physiology, The Islamia University of Bahawalpur (IUB), Pakistan (for lab work). Cholistan desert is located at latitudes 27˚42´and 29˚45´North and longitudes 69˚52´and 75˚24´East and at an altitude of 112m above the sea level. The climate of this area is arid, hot subtropical and monsoonal with the average annual rainfall of 180 mm. The mean annual temperature is 28.33˚C, with the month of June being the hottest when the daily maximum temperature normally exceeds 45˚C [12, 13].

### Ethics statement

The research work was approved by the Departmental Research Ethics Committee, Department of Physiology, The Islamia University of Bahawalpur, Pakistan vide Letter No PHYSIO-1/2022-188 dated 05-10-2022.

### Experimental animals

Cholistani camel herds being reared by the nomadic pastoralists in Cholistan desert were randomly selected for incorporation in the study. Animals were selected randomly and sampled

irrespective of their age and sex. Males and female camels were considered adults if above the age of 3 and 4 years, respectively. All the animals were being reared under similar management and feeding conditions of pastoralism either under transhumanie or nomadic pastoral livestock production systems [14]. Rotational-herding is normally exercised for camel herds by the Cholistani pastoralists, according to which the camel herds move around in the desert under watchful eyes of the shepherd for first three months. Later on, for the next nine months they are left alone to graze and move as per their will. However, the animals never leave their area or their herds [14]. This is somewhat different to the split-herding used for Cholistani cattle [12]. From a total of three randomly selected camel herds, apparently healthy camels (n = 215) were selected for the study. The general health status of animals was ascertained through a thorough anamnesis from the livestock owners and clinical signs. The animals which were found to be lethargic, depressed, off-feed and segregated from the herd (as per the anamnesis taken from the pastoralist herders) were not included in the study.

## Blood collection

The blood sampling was conducted from July to October, 2022. About 5 mL blood was collected aseptically in anticoagulant-added tubes (0.5 M EDTA) with the help of a disposable syringe from the high neck jugular vein of each animal. The same restraining technique with same personnel and time (0900 AM) were used to minimize the stress in animal and also to normalize blood collection procedure. The blood samples were mixed by gentle inversion and transported in an ice box to the Physiology Post-graduate Lab, IUB, Pakistan, refrigerated and analyzed within 24h for hematological analyses. Each animal was bled once only.

## Hematological analyses

The blood samples were analyzed for PCV and Hb as per the protocols prescribed by the WHO and in vogue, and are considered as gold standard tests for PCV and Hb determination, respectively [1]. PCV was deduced by microhematocrit centrifuge method using microcentrifuge (Sigma Aldrich, Model 5254, Germany) and reading as percentage (%) was taken through a hematocrit card-reader. The reading was used for calculating Hb as its third and was dubbed as Hemoglobin Calculated (HbC).

   The Hb concentration was also determined through Drabkin's reagent (HbD) using cyanmethaemoglobin method and a commercial Hb Kit (AMP Diagnostics, BD6100-E V4.0-CE Ameda Labordiagnostik GmbH, Germany) [15]. The Hb was calculated as per formula prescribed in instructions manual.

## Statistical analyses

Statistical Package for Social Science (SPSS for Windows version 12, SPSS Inc., Chicago, IL, USA) was used for data analysis. Means (±SE) and 95% CI for hematological attributes (PCV and Hb) were computed using prescribed formulae. For the purpose of analyses, considering the non-normal nature of the attained data, the Mann Whitney-U test was implied as a nonparametric test for deducing difference between HbD and HbC, and between HbD and corrected Hb (CHb) for all study groups (young = 85, adult = 130; females = 121, males = 94).

   Scatterplots were drawn and linear regression was carried out between the following in order to deduce regression prediction equations [9, 16]:

a.  HbD and PCV

b.  HbD and HbC

**Table 1. Mean (±SE) values and confidence intervals for hemoglobin determined (HbD), hemoglobin calculated (HbC) and Packed Cell Volume in Cholistani camels (n = 215).**

| Groups | | HbD (g/dL) | | HbC (g/dL) | | Sig | PCV (%) | |
|---|---|---|---|---|---|---|---|---|
| | | x±SE | CI | x±SE | CI | | x±SE | CI |
| **Gender** | Females (n = 121) | 10.8±0.2 | 10.4–11.2 | 9.4±0.2 | 9.0–9.8 | 0.01* | 28.3±0.6 | 27.2–29.5 |
| | Males (n = 94) | 11.2±0.3 | 10.7–11.8 | 10.8±0.4 | 10.0–11.7 | 0.04* | 32.6±1.2 | 30.1–35.1 |
| **Age** | Young (n = 85) | 11.0±0.3 | 10.3–11.6 | 9.3±0.5 | 8.2–10.2 | 0.005* | 27.8±1.4 | 24.8–30.7 |
| | Adult (n = 130) | 11.1±0.2 | 10.7–11.5 | 10.4±0.3 | 9.8–11.0 | 0.05* | 31.3±0.8 | 29.6–32.9 |
| **Overall (n = 215)** | | **11.0±0.1** | **10.7–11.4** | **10.2±0.3** | **9.7–10.7** | **0.007*** | **30.7±0.7** | **29.2–32.1** |

*Significant at P≤0.05 within rows for each group between hemoglobin determined and hemoglobin calculated.

HbD = Hemoglobin determined spectrophotometrically; HbC = Hemoglobin calculated as 1/3$^{rd}$ of PCV; PCV = Packed cell volume

c. The difference of HbD and CHb (HbD-CHb), and means of measurements (HbD+CHb/2)

Regression prediction equations were used for calculating CHb. Level of agreement between HbD and CHb was assessed through Bland Altman Agreement Analysis.

## Results

In the present study, hemoglobin determined spectrophotometrically (HbD) and hemoglobin calculated (HbC) as one third of Packed Cell Volume (PCV) was assessed for statistical difference at P≤0.05. Furthermore, PCV conducted through microhematocrit method was studied for interrelationship between the HbD and corrected hemoglobin (HbC) (through a formula attained by regression analyses).

Regarding normality of studied attributes (HbD, PCV and HbC), the Shapiro-Wilk test revealed that all the three studied attributes were not distributed normally.

Mean (±SE) values and 95% CI for hematological attributes (HbD, HbC and PCV) in Cholistani camels (n = 215) are presented in Table 1. The mean (±SE) values for HbD and PCV were within the normal reference ranges reported for camels. The overall results indicated a significant (P≤0.05) difference between HbD and HbC. Similar results were attained for all study groups (females *vs* males, and adults *vs* young) of the present study.

The results for linear regression for all study groups are presented in Table 2. Significantly (P≤0.01) higher positive correlation coefficient was noticed for young camels (r = 0.830; adjusted r-square = 0.68) between HbD and PCV, and between HbD and HbC.

Regression equations were developed to validate the 1/3$^{rd}$ association between PCV and HbD for all study groups. The regression equation of overall data hence attained *i.e.* Hb (g/d) = 0.18(PCV)+5.4 was used to deduce Hb dubbed as corrected Hb (CHb). A non-significant

**Table 2. Linear regression between various hematological attributes for Cholistani camels (n = 215).**

| Groups | | HbD vs PCV | HbD vs HbC | r | Adjusted r Square |
|---|---|---|---|---|---|
| **Gender** | Females (n = 121) | y = 0.23; x+4.3 | y = 0.7; x+4.3 | 0.654* | 0.41 |
| | Males (n = 94) | y = 0.20; x+5.3 | y = 0.54; x+5.3 | 0.79* | 0.62 |
| **Age** | Young (n = 85) | y = 0.19; x+5.8 | y = 0.60; x+5.8 | 0.830* | 0.68 |
| | Adult (n = 130) | y = 0.19; x+5.3 | y = 0.60; x+5.3 | 0.751* | 0.56 |
| | **Overall (n = 215)** | **y = 0.18; x+5.4** | **y = 0.55 x+5.5** | **0.753*** | **0.56** |

*Significant correlation at P≤0.01.

HbD = Hemoglobin determined spectrophotometrically; HbC = Hemoglobin calculated as 1/3rd of PCV; PCV = Packed cell volume

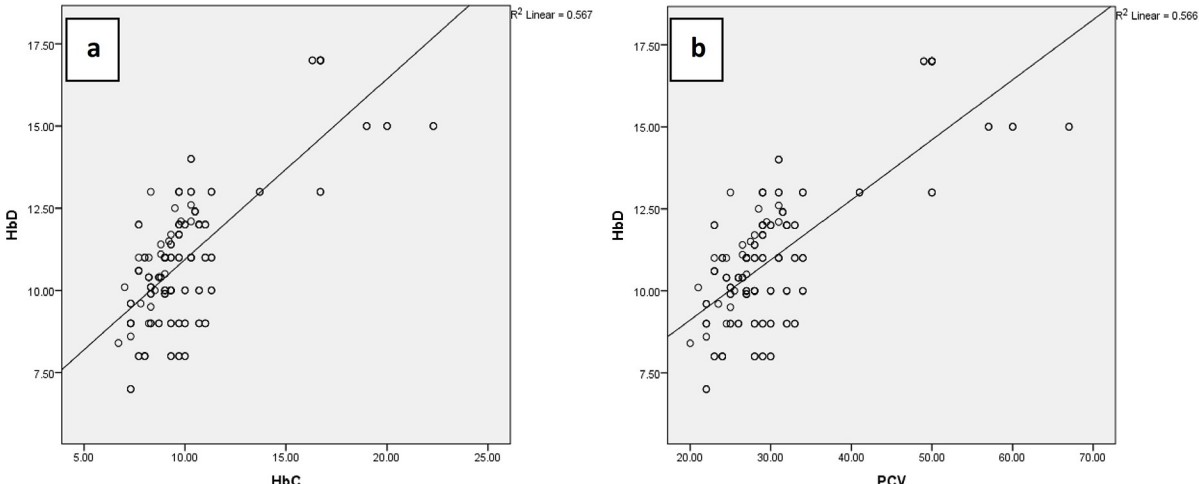

**Fig 1.** Scatterplot for logilinear regression between a) hemoglobin determined spectrophotometrically (HbD) and Packed Cell Volume (PCV) (n = 215; r = 0.753), and b) between hemoglobin determined spectrophotometrically (HbD) and hemoglobin calculated as one-third of Packed Cell Volume (HbC) (n = 215; r = 0.753) in Cholistani camels.

(P≥0.05) difference was noticed between HbD and CHb. This equation is therefore, considered valid for deducing Hb from PCV in all age and gender groups of Cholistani camels.

The scatterplots of spectrophotometrically determined Hb (HbD) versus PCV, and HbD versus hemoglobin calculated as 1/3rd of PCV (HbC) have been given in Fig 1A and 1B. Similarly, the scatterplots and Bland and Altman chart for difference between HbD and CHb (HbD-CHb) versus average of HbD and CHb (HbD+CHb/2) is given in Fig 2. Satisfactory agreement between HbD and CHb was noticed and the data was distributed close to the mean difference line (Mean = 0.1436, 95%CI = 3.00–2.72).

## Discussion

Cattle, sheep, goats and camels are the predominant types of livestock and the total population of livestock in Cholistan desert was estimated to be 12, 09, 528 heads, comprising of 47% cattle, 30% sheep, 22% goats and 1% camels [12, 17]. In tropical pastoral system, in addition to shortage and poor quality of foodstuff, the decrease in the productive and reproductive potential of livestock can also be ascribed to the incidence of infections such as helminthiasis, trypanosomiasis, theileriosis, tick burden and tick borne infectivity. The parasitic infestation is one of the most important reasons of disease and production loss in the livestock by causing anemia and mostly death in heavily infected animals [18–21]. Normally, for the diagnosis of anemia, PCV and Hb levels of the blood picture/complete blood count (cbc) are considered valid enough. In resource-poor settings (such as in Pakistan), Automated Veterinary Hematology Analyzers are scanty. And human blood analyzers are usually being used for blood of livestock [22]. This poses the threat of erroneous results as the human analyzers are differently validated than the Veterinary Hematology Analyzers [23]. This endorses the vitality of gold standard tests such as microhematocrit method (for PCV) and cyanmethemoglobin method (for Hb levels) [4, 15]. The present study included these two gold standard methods for deducing PCV and Hb in camel blood, and after attaining appropriate interrelationship between these attributes, puts forth a simple, pen-side hematological formula of Hb (g/dL) = 0.18(PCV)+5.4 for estimating Hb from PCV.

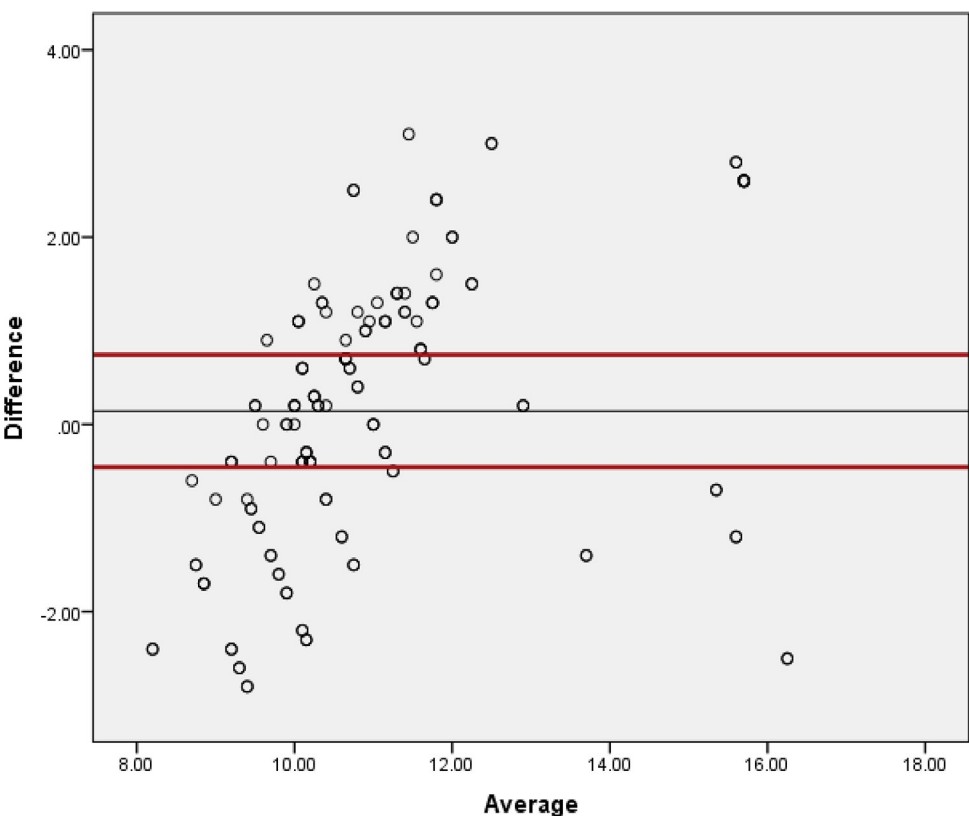

**Fig 2. Scatterplot of Bland and Altman test between difference of hemoglobin determined spectrophotometrically and corrected hemoglobin (HbD-CHbC) and average of both hemoglobins (HbD+CHb/2) in Cholistani camels (n = 215).** Black line indicates mean difference (0.1436) whereas the upper and lower red lines indicate upper (3.00) and lower (-2.72) values for 95% CI.

Considering the 'hematological rule of three' which is being implied in human medical practice, it has been well elucidated that Hb can be estimated as $1/3^{rd}$ of the PCV for apparently healthy human populations having normocytic normochromic erythrocytes [2, 24, 25]. On similar grounds, some studies on human blood have also negated the validity of this convention. In a malaria-endemic setting, this convention was not found valid in children and it was concluded that age, gender, season of sampling and physiological status of humans affects relationship between Hb concentration and PCV [7]. It was hence dubbed impossible to deduce a validated mathematical formula for their relationship as shown by other studies as well [6]. The earlier study dates back to 1994 which was conducted on human blood and endorsed that Hb was accurately measured as $1/3^{rd}$ of PCV and vice versa [26].

Regarding veterinary medical sciences, scanty work has been reported for such hematological validations and formulae. The pioneering work on the interrelationship of PCV and Hb in livestock has been conducted for indigenous African cattle breeds and a formula of 0.28(PCV) +3.11 has been reported for Hb estimation in g/dLs [9]. A research work on various avian orders noticed that reasonable approximates of Hb levels can be obtained through the determination of PCV or *vice versa*. A simple correlation of Hb = 0.30×PCV, was put forth as a sound calculation of Hb from the relevant PCV for blood samples from the orders Anseriformes, Columbiformes, Falconiformes, Galliformes, Passeriformes, Psittaciformes, Sphenisciformes, and Strigiformes. While in case of Phoenicopteriformes, a separate regression equation; Hb = 0.217×PCV+6.69, has been provided as a reasonable estimate of Hb from determined

PCV [27]. Very recently, our laboratory has reported pen-side hematological formulae for Hb estimation in goat blood [10] and for the blood of Cholistani breed of cattle [11]. This is the first report on such interrelationship for camels being reared under pastoralism in Cholistan desert of Pakistan. In the present study, in order to assess level of agreement, Bland-Altman test was implied which is considered a gold standard test for testing the outcome of two techniques. Results of comparison between the data of HbD and CHB of the present study indicate a substantially strong level of agreement as the values of upper and lower CIs are close to the mean distribution line. These results are also strengthened by the results of correlation coefficient as the r-value in the present study is 0.753.

## Conclusions

Summing up, a convention of human clinical medicine that Hb concentration is a $1/3^{rd}$ of PCV and vice versa cannot be implied for the camels. However, a different equation *i.e.* Hb (g/dL) = 0.18(PCV)+5.4 may provide reliable results for Hb estimation from the PCV in this specie. The results of the study may be of substantial value to the researchers, academicians and veterinary clinicians of resource-poor areas. It is suggested that other mathematical formulae regarding hematological attributes being used in human clinical medicine may also be validated for various use in veterinary medical practice.

## Acknowledgments

The authors are grateful to Dr. Muhammad Sohail Khan, Director Livestock Cholistan, Livestock & Dairy Development Department (L&DD), South Punjab, Pakistan for his continued support and guidance for this research work. Gratitude is also extended towards the dedicated technical staff and field veterinarians of the L&DD, South Punjab, Pakistan for their assistance in field visits of Cholistan desert and sample collection.

## Author Contributions

**Conceptualization:** Umer Farooq.

**Data curation:** Sajid Hameed.

**Formal analysis:** Nouman Sajjad, Shahbaz Ahmad.

**Investigation:** Musadiq Idris, Haroon Rashid.

**Methodology:** Zia Ur Rehman.

**Supervision:** Aisha Mahmood.

**Writing – original draft:** Mushtaq Hussain Lashari.

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
