## [Decision Letter · Decision Letter 0]

1 Mar 2023

PONE-D-22-35645Assessing the potential of packed cell volume for deducing hemoglobin: Cholistani camels in perspectivePLOS ONE

Dear Dr. Lashari,

Thank you for submitting your manuscript to PLOS ONE. After careful consideration, we feel that it has merit but does not fully meet PLOS ONE’s publication criteria as it currently stands. Therefore, we invite you to submit a revised version of the manuscript that addresses the points raised during the review process.

We look forward to receiving your revised manuscript.

Kind regards,

Muhammad Mazhar Ayaz, Ph.D

Academic Editor

PLOS ONE

Journal Requirements:

“The authors are thankful to the ‘Pakistan Science Foundation’ for provision of research fund under research grant No. PSF/NSLP/P-IUB-931 titled “Devising and Validating Pen-side 10 Hematological Tests as an Enhanced Approach to the Diagnosis of Anemia in Cholistani Livestock (Camels and Cattle)”.”

“the authors received funding/award”

“the authors received funding/award”

“I have read the journal's policy and the authors of this manuscript have the following competing interests

it will helpful to manage the health of animals, and also helpful for professionals”

Reviewers' comments:

Reviewer's Responses to Questions

**Comments to the Author**

1. Is the manuscript technically sound, and do the data support the conclusions?

Reviewer #1: Partly

Reviewer #2: Yes

2. Has the statistical analysis been performed appropriately and rigorously? 

Reviewer #1: Yes

Reviewer #2: Yes

3. Have the authors made all data underlying the findings in their manuscript fully available?

Reviewer #1: Yes

Reviewer #2: No

4. Is the manuscript presented in an intelligible fashion and written in standard English?

Reviewer #1: No

Reviewer #2: Yes

5. Review Comments to the Author

Reviewer #1: Manuscript evaluation (ID PONE-D-22-35645)

Overall assessment

I have reviewed a research article entitled ‘Assessing the potential of packed cell volume for deducing hemoglobin: Cholistani camels in perspective)’ submitted to PLOS ONE by Farooq et al. Overall the title of the manuscript is useful in context of monitoring the health of the fish. In general, the purpose and procedure of the experiment and the experimental data are written clearly and performed through appropriate experimental methods, which supporting a coherent conclusion in the study. However, there are several questions, so I suggest a major overhaul of the MS before its further consideration.

Under Abstract

The section is well written, however, the statement “A good agreement between HbD and CHb and there was no proportional bias on the distribution of data around the mean difference line (Mean= 0.1436, 95%CI= 3.00—2.72).” needs to be rephrased.

Under Introduction

Comments

The first few lines of the introductory paragraph seem to simply define the PCV which is not required, rather the authors must through light on the functional role of PCV and its existing relationship with Hb with reference to the species under study.

Authors must cite the research showing the existing relationship of PCV and Hb as deduced in human and other animal subjects.

Under Methodology

The methodology employed is robust, however information about following points is lacking Time of blood collection, type and volume of syringe used for blood collection, type of anti-coagulant used, storage temperature etc.

The sample size is ambiguous. Have you sub classed the data in terms of gender and age? How many replications were performed?

Under statistical analysis

As per the authors, the data did observe the normal distribution and use of non-parametric tests is understood. However, when the going for regression analysis, have you gone for data transformation? and what type of post hoc test was performed?

Under results

Comments

Table titles must be appropriate. Make sure, the use of uniform terminology across the manuscript. Use either Colistani camels or cattle in both of the tables 1 and 2.

In table 2, the information is little and needs more clarification. Authors must add the equations of both the regression analysis performed (HbD vs PCV and HbD vs HbC) and later on show the changes in adjusted r2.

Under discussion

The discussion is written very poorly and needs thorough revision. Authors should take into consideration the following key points

Specific comments

The first paragraph of the discussion section is giving information which is not supported by the results of the study. You should limit such information to the introduction portion.

There is a very little information about the hematology of camels in the discussion section so the authors must include good number of comparisons citing the previous reports on other camel species.

Results of the study have been very poorly discussed. The authors must discuss their results, and provide valid grounds such as intrinsic physiological/species- specific, gender, age, body size, basis of the significant differences observed. Furthermore the study needs validation in terms of the statistical methodology which is being recommended and widely used in the studies with similar reports.

There are several contradictory statements in the discussion. On the one hand authors indicate that there are reports about high infection of cattle in the region of study which limit the power of PCV to predict the accurate Hb content, as you have cited about malaria settings in human medicine. But on the other hand the discussion does not give any idea about whether the formula that is being endorsed here has some kind of operator which undoes the effect of disease, particularly in cases where PCV might not get significantly altered.

The overall discussion needs to revise thoroughly to make it scientifically convincing and additionally there are several spelling mistakes, throughout the discussion section which should be corrected.

In my opinion that authors should reframe the manuscript as per suggestions and quarries raised in the manuscript. Presentation needs to be improved in the methodology results and discussion sections. The manuscript should be re-considered for publication after major revision.

Reviewer #2: The manuscript, submitted as a technical note, “Assessing the potential of packed cell volume for deducing hemoglobin: Cholistani camels in perspective” by Dr. Lashari and colleagues presents a methodology to estimate hemoglobin concentration, based on PCV values for camels.

The formula proposed: Hb concentration (g/dL)= 0.18(PCV)+5.4 could be very useful for a field evaluation of anemia, mainly for camels, that have a more limited source of references.

Initially, since one of the goals was to develop tools to improve Cholistani camels health and productivity, I would suggest to explore a little bit more about the camels in the manuscript. For instance: are the results found for PCV values in the normal range for camels? Was the higher correlation coefficient between HbD and PCV found for young camels expected? Many references from the discussion are from cattle. It would be nicer to have more information about camels in the discussion. Otherwise, this manuscript will be very similar to the ones published by the group (Ahmad et al. 2022a and Ahmad et al. 2022b). The authors should consider rewriting some parts, to avoid self-plagiarism.

Other doubt is about the term: “pen-side”. Usually, when we use “pen-side” analysis is for something that could be done at the moment an animal is examined. In this case, since it is necessary to go to a laboratory (or at least, to have the hematocrit done) to estimate the hemoglobin concentration, can we say it is pen-side?

Other issues:

Materials and methods

More information about the camels could be given, like: are they from a specific breed? What are they purpose? The camels were from how many herds?

It would also be nice to explain the age groups. When is a camel considered adult? With what age?

“All the animals were being reared under similar management and feeding conditions of pastoral- ism. Split-herding is normally exercised for livestock by the pastoralists, according to which the young ones (calves in this case) are kept at their pens near the “Tobas” (natural or man-mad water reservoirs of the desert), while the adults are sent for grazing till night-time (Farooq et al., 2010).”

Is this about the camels? Because it is the same explanation in the cattle article.

Results

Table 2 –Cholistani Camels?

Discussion

….the livestock population is on the boom with the succession of years (Farooq et al., 2010). Maybe here we need recent reference, that could confirm this boom in livestock populations.

The parasitic infestation is one of the most important reasons of disease and production loss in the livestock by causing anemia and mostly death in heavily infected animals (Grace et al., 2007).

This is a reference for cattle. It would be interesting to have more references like this, but for camels.

References:

Ahmad, S., Farooq, U., Lashari, M.H., Idris, M., Ur-Rehman, Z., Khan, N. & Sajjad, N. (2022). Devising and validating a pen-side hematological formula for hemoglobin estimation in Cholistani cattle. Tropical Animal Health and Production, 54, pp.1-6.

It should be (2022b)

6. PLOS authors have the option to publish the peer review history of their article (what does this mean?). If published, this will include your full peer review and any attached files.

Reviewer #1: **Yes: **Imtiaz Ahmed

Reviewer #2: No

---

## [Author Response · Author response to Decision Letter 0]

28 Mar 2023

The comments/suggestions by the Editor and both the reviewers have been thoroughly incorporated in the newly amended draft, please.

---

## [Editor Report · Decision Letter 1]

29 Mar 2023

Investigating the potential of packed cell volume for deducing hemoglobin: Cholistani camels in perspective

PONE-D-22-35645R1

Dear Dr. Lashari,

We’re pleased to inform you that your manuscript has been judged scientifically suitable for publication and will be formally accepted for publication once it meets all outstanding technical requirements.

Kind regards,

Muhammad Mazhar Ayaz, Ph.D

Academic Editor

PLOS ONE
---

## [Editor Report · Acceptance letter]

3 May 2023

PONE-D-22-35645R1 

Investigating the potential of packed cell volume for deducing hemoglobin: Cholistani camels in perspective 

Dear Dr. Lashari:

I'm pleased to inform you that your manuscript has been deemed suitable for publication in PLOS ONE. Congratulations! Your manuscript is now with our production department. 

Kind regards, 

on behalf of

Dr. Muhammad Mazhar Ayaz 

Academic Editor

PLOS ONE